# Superinfection exclusion: A viral strategy with short-term benefits and long-term drawbacks

**Michael Hunter** *, **Diana Fusco** *

Cavendish Laboratory, University of Cambridge, Cambridge, United Kingdom

* mth47@cam.ac.uk (MH); df390@cam.ac.uk (DF)

## Abstract

Viral superinfection occurs when multiple viral particles subsequently infect the same host. In nature, several viral species are found to have evolved diverse mechanisms to prevent superinfection (superinfection exclusion) but how this strategic choice impacts the fate of mutations in the viral population remains unclear. Using stochastic simulations, we find that genetic drift is suppressed when superinfection occurs, thus facilitating the fixation of beneficial mutations and the removal of deleterious ones. Interestingly, we also find that the competitive (dis)advantage associated with variations in life history parameters is not necessarily captured by the viral growth rate for either infection strategy. Putting these together, we then show that a mutant with superinfection exclusion will easily overtake a superinfecting population even if the latter has a much higher growth rate. Our findings suggest that while superinfection exclusion can negatively impact the long-term adaptation of a viral population, in the short-term it is ultimately a winning strategy.

**Data Availability Statement:** The code used throughout this work has been deposited on

## Author summary

Viral social behaviour has recently been receiving increasing attention in the context of ecological and evolutionary dynamics of viral populations. One fascinating and still relatively poorly understood example is superinfection or co-infection, which occur when multiple viruses infect the same host. Among bacteriophages, a wide range of mechanisms have been discovered that enable phage to prevent superinfection (superinfection exclusion) even at the cost of using precious resources for this purpose. What is the evolutionary impact of this strategic choice and why do so many phages exhibit this behaviour? Here, we conduct an extensive simulation study of a phage population to address this question. In particular, we investigate the fate of viral mutations arising in an environment with a constant supply of bacterial hosts designed to mimic a "turbidostat," as these are increasingly being used in laboratory evolution experiments. Our results show that allowing superinfection in the long-term yields a population which is more capable of adapting to changes in the environment. However, when in direct competition, mutants capable of preventing superinfection experience a very large advantage over their superinfecting counterparts, even if this ability comes at a significant cost to their growth rate. This indicates that while preventing superinfection can negatively impact the long-term prospects of a viral population, in the short-term it is ultimately a winning strategy.

GitHub at: https://github.com/FuscoLab/phage_coinfection.

**Funding:** MH acknowledges studentship funding from EPSRC under grant number EP/R513180/1. This work was performed using resources provided by the Cambridge Service for Data Driven Discovery (CSD3) operated by the University of Cambridge Research Computing Service, provided by Dell EMC and Intel using Tier-2 funding from the Engineering and Physical Sciences Research Council (capital grant EP/P020259/1), and DiRAC funding from the Science and Technology Facilities Council. The funders had no role in study design, data collection and analysis, decision to publish, or preparation of the manuscript.

**Competing interests:** The authors have declared that no competing interests exist.

## Introduction

Bacteriophages (phages) are viruses that infect and replicate within bacteria. Much like many other viruses, reproduction in lytic phage is typically characterised by the following key steps: adsorption to a host cell, entry of the viral genetic material, hijacking of the host machinery, intracellular production of new phage, and finally, the release of progeny upon cell lysis. Phages represent one of the most ubiquitous and diverse organisms on the planet, and competition for viable host can lead to different strains or even species of phage superinfecting or co-infecting the same bacterial cell, ultimately resulting in the production of more than one type of phage (Fig 1a) [1–3]. In the following, we define infection terminology in line with Turner & Duffy [4], such that co-infection occurs when two or more phage have successfully infected a single bacteria, and superinfection occurs when there is a delay between infection by the first and second phage. Therefore, all cells which have been successfully superinfected can be said to be co-infected [4]. To account for different usages throughout the literature and across fields, we also refer to multiple infections, to indicate any case where multiple viruses exist within a single host simultaneously.

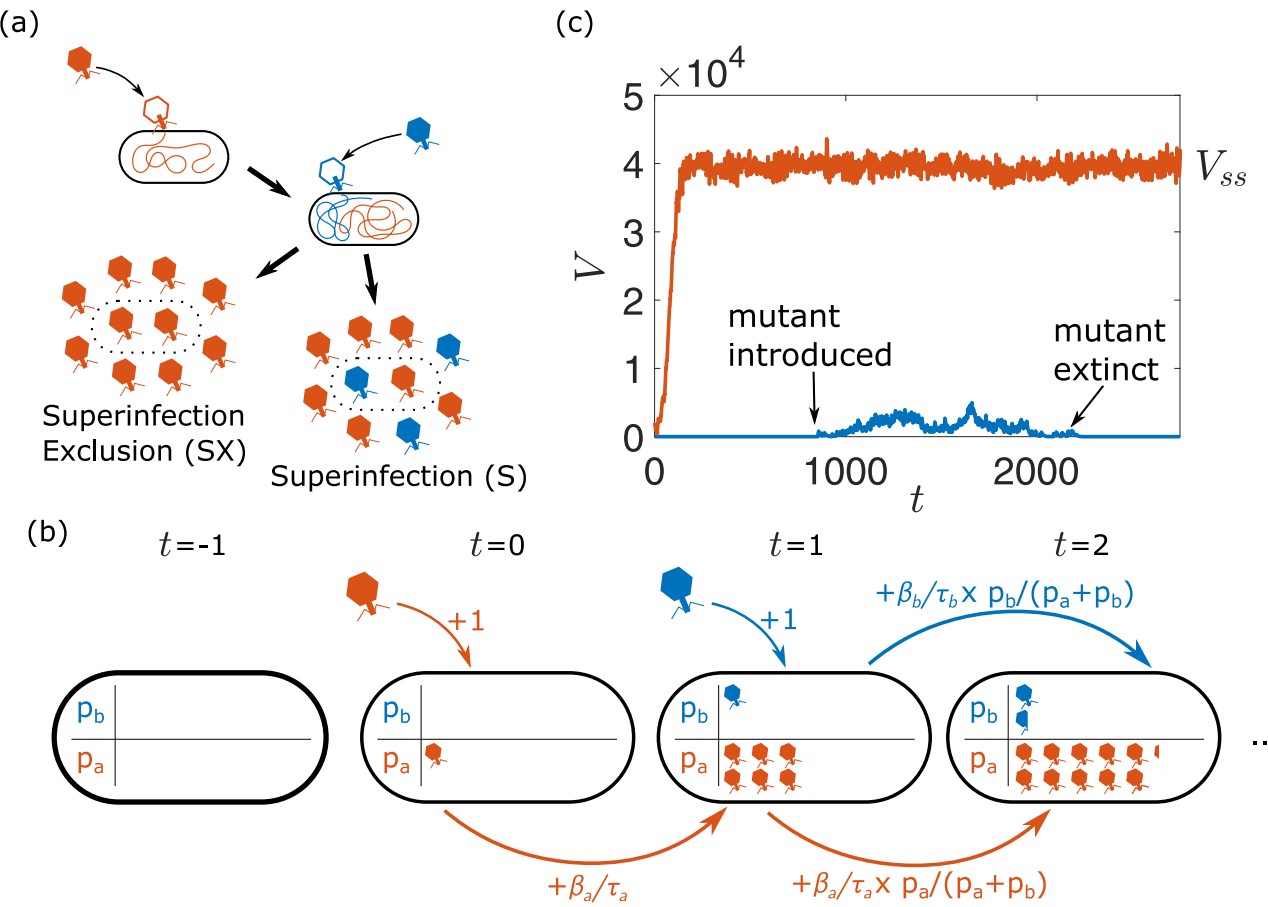

**Fig 1. Modelling setup.** (a): In superinfection-excluding scenarios, all of the progeny released as the cell lyses are copies of the initial infecting phage, whereas when superinfecting is permitted, the progeny are split between both types of phage. (b): During superinfection, pseudo-populations $p_a$ and $p_b$ are used to represent the growth of phage inside the host cells. These populations increase by 1 whenever a phage infects the host, and each population increases by some fraction of its rate $\beta/\tau$ determined by the relative size of the populations in the previous step. (c): An example realisation of the simulation. The resident phage population initially grows until it reaches a steady state, at which point a mutant phage is introduced to the population, and the simulation is run until extinction or fixation of the mutant.

Interestingly, several phages have evolved mechanisms that prevent superinfection (superinfection exclusion). This can be achieved at the early stage of infection, by preventing further adsorption of phage, or at a later stage, by preventing the successful injection of subsequent phage DNA [5, 6]. For instance, bacteriophage T5 encodes a lipoprotein (Llp) that is synthesised by the host at the start of infection and prevents further adsorption events by blocking the outer membrane receptor site (FhuA protein) [7, 8]. Bacteriophage T4 encodes two proteins, Imm and Sp, that prevent superinfection by other T-even phages by inhibiting the degradation of bacterial peptidoglycan, whose presence hinders the DNA transfer across the membrane [9, 10].

Given that populations which allow and prevent superinfection both exist in the wild, it is natural to wonder what impact either strategy has on the evolution of viral populations. This question has been studied in various systems from the perspective of intracellular interactions and competition [11–19]. Multiple infections allow for the exchange of genetic material between viruses through recombination, which can increase diversity and improve the efficiency of selection, but may also decrease fitness by promoting the presence of deleterious mutants at low frequencies [20–22]. Additionally, in RNA viruses with segmented genomes, multiple infections can lead to hybrid offspring containing re-assorted mixtures of the parental segments (reassortment). This mechanism can in principle improve selection efficiency, as reassorted segments may generate highly deleterious variants that will be easily out-competed by the rest of the population [23]. Multiple infections can also lead to viral complementation, where defective viruses can benefit from superior products generated by ordinary viruses inside the host [23–27]. This process increases the diversity of the population, but also allows cheating individuals to persist in the viral population for long times [23, 24].

The likelihood of multiple infections occurring increases with the number of free phage available per viable host—multiplicity of infection (MOI)—and several experimental systems have been used to study the impact of MOI on viral dynamics [25, 26, 28–32]. For instance, high MOI in RNA phage $\phi 6$ has been shown to result in a behaviour conforming to the Prisoner's Dilemma strategy in game theory, and a reduction in viral diversity [28–31, 33]. Theoretically, the same question has been investigated in different scenarios [34], in particular in the context of human immunodeficiency virus (HIV) infections [20, 21, 35–40]. These studies have focused on determining whether multiple infections preferentially occur simultaneously or sequentially, in an effort to explain experimental data, and on the role of recombination in the acquisition of drug resistance, showing that its impact depends on the effective population size. The role of MOI has also been studied in terms of diversity and evolution of the viral population [20, 21, 37, 41–46], with theoretical predictions suggesting that multiple infection favours increased virulence, and that within-host interactions can lead to a more diverse population.

Despite the active work in the area, several fundamental questions on the role of superinfection exclusion on viral dynamics remain unanswered. First, while decreasing MOI in viral populations that allow superinfection decreases the *likelihood* of superinfection, it does not introduce a superinfection exclusion mechanism that prevents superinfection altogether, making it difficult to draw conclusions about the (dis)advantages of this viral strategy. Second, little is known about how the occurrence of superinfection alone, before even accounting for the additional effects of any intracellular interactions, impacts the evolution of viral populations, particularly when it comes to fundamental evolutionary outcomes such as mutant fixation probabilities. A quantitative understanding of this baseline behaviour is necessary to evaluate the impact of the many additional intracellular interactions that can occur (recombination, defective viruses, etc.). The limited work in this area has shown that in the absence of intracellular interactions, high MOI in superinfecting viral populations can promote the presence of

disadvantageous mutants in the "short term," and obstruct it in the "long term" [47, 48], but how the evolutionary outcomes in each case depend on the parameters describing the viral life-cycle (adsorption rate, lysis time and burst size) and the (dis)advantages of either strategy remain unclear.

Here, we explore how allowing or preventing superinfection impacts the evolutionary fate of neutral and non-neutral variants in a simulated well-mixed phage population with constant, but limited, availability of host. We choose to focus on superinfection exclusion mechanisms that allow secondary adsorption events, but prevent DNA insertion, so that in isolation the phage growth dynamics is the same in the two cases and a direct comparison between the (dis)advantages of the two strategies is more straightforward. We first quantify the effective population size of superinfecting (S) and superinfection-excluding (SX) populations to estimate how these strategies affect genetic drift. We then turn our attention to the effect of non-neutral mutations on (i) the phage growth rate in isolation and (ii) their ability to out-compete the wild-type. Having characterised both the neutral dynamics and the fitness of different variants, we put both aspects together to explore the balance between drift and selection in superinfecting and superinfection-excluding populations, showing that selection is consistently more efficient in superinfecting populations. Finally, we study the evolutionary fate of a mutation which changes whether an individual is capable of preventing superinfection or not. Overall, this work establishes a baseline expectation for how the simple occurrence of superinfection impacts fundamental evolutionary outcomes and provides insights into the selective pressure experienced by viral populations with limited, but constant host density.

## Results

### Computational modelling framework

We study the evolutionary fate of phage mutants using a stochastic agent-based model. We simulate a well-mixed population of phages $V$ interacting with a population of host bacteria that is kept at a constant density, similarly to a turbidostat [49, 50]. Each phage has a defining set of life history parameters, namely an adsorption rate $\alpha$, a lysis time $\tau$ and a burst size $\beta$, and each bacteria can either be in an uninfected $B$ or an infected $I$ state.

In each simulation time-step, adsorption, phage replication within the host and lysis occur. The number of infecting phage $V_I$ in each step is drawn from a Poisson distribution whose mean corresponds to the expected value $\alpha V(B + I)$ in a well-mixed population. The infecting phage are removed from the pool of free phage, and $V_I$ bacteria, whether infected or uninfected, are chosen uniformly and with replacement to be the infection target. In both superinfecting and superinfection-excluding scenarios, the final lysis time $\tau$ of the host is set by the first phage to infect it and it is treated as deterministic to limit the number of model parameters. This choice was made for the sake of simplicity, given the complex and varied nature of superinfection mechanisms [1–3]. A preliminary analysis of the effect of stochasticity in lysis time is presented in S1 Appendix. In the case where multiple phage infect the same host in a single time-step, the 'first' phage is chosen uniformly among those infecting the host. Phage replication within the host post-adsorption depends on whether superinfection is allowed or prevented:

**Absence of superinfection.** $\tau$ steps after the first adsorption event, the bacteria will lyse, releasing new phage into the pool of free phage. The number of phage released $Y$ is drawn from a Poisson distribution with mean $\beta$.

**Presence of superinfection.** Pseudo-populations tracking the growth of phage inside the host are used (see Fig 1b). Because here we focus on the case of two superinfecting phage

populations, this results in two pseudo-populations $p_a$ and $p_b$. During the intermediate steps between the first adsorption event and lysis, in the case where there is only one type of phage inside the host, that population will grow at a constant rate $\beta/\tau$, where $\beta$ and $\tau$ are both specific to the type of phage (i.e. $p_a$ grows at rate $\beta_a/\tau_a$ and $p_b$ grows at rate $\beta_b/\tau_b$). This is to reflect previous reports of a positive linear relationship between lysis time and burst size [51]. In the event where both types of phage are present within the host, to reflect the intracellular competition for the host's resources, each population increases by only a fraction of its potential $\beta/\tau$ determined by the size of each population at that time, i.e. $p_a$ increases by an amount $\beta_a/\tau_a \times p_a/(p_a + p_b)$ and $p_b$ increases by an amount $\beta_b/\tau_b \times p_b/(p_a + p_b)$. At the point of lysis, the total number of phage released $Y$ is drawn from a Poisson distribution with mean $p_a + p_b - V_n$, where $V_n$ represents the number of viruses that infected the host prior to lysis. This is to ensure that, in the event where a cell is only infected by 1 type of phage, its mean burst size remains $\beta$, regardless of how many phages had infected the cell until that point. The number of phage released of one type $Y_a$ is then drawn from a binomial distribution with $Y$ attempts and probability $p_a/(p_a + p_b)$ of success, with any remaining phage being the other type ($Y_b = Y - Y_a$).

Following lysis, the lysed bacteria are immediately replaced with a new, uninfected host, resulting in a bacterial population of constant size. We also introduce a decay, or removal, of free phage at rate $\delta$, which accounts for natural phage decay and the outflow of the turbidostat system.

Simulations were initialised with $B_0$ uninfected bacteria and $2B_0$ "resident" phage, and then run until the phage, uninfected bacteria and infected bacteria populations each reached steady state values ($V_{ss}$, $B_{ss}$ and $I_{ss}$ respectively), as determined by their running average (Fig 1c). This steady state arises due to a balance between phage production and loss and it is independent of the initial number of phages (S1 Fig).

## Superinfection leads to a larger effective population size

First, we find that genetic diversity consistently declines faster in populations that prevent superinfection, indicating a smaller effective population size $N_e$ when compared to superinfecting populations (see Methods). This can be intuitively understood by considering that in the superinfecting scenario, each phage has more opportunity to successfully infect a host cell, since secondary infections can result in the production of some offspring when the cell lyses. Therefore, more phage are able to contribute to the next generation, thereby slowing down diversity loss.

In addition, Fig 2 shows that in both superinfecting and superinfection-excluding populations higher adsorption rate and burst size, and shorter lysis time result in larger effective populations. This observation is, however, partially attributable to the change in total phage population $N_T = (V_{ss} + \beta I_{ss})$, where $V_{ss}$ indicates the steady state free phage population, $I_{ss}$ indicates the steady state number of infected bacteria, and so $\beta I_{ss}$ represents the number of phage that inevitably will join the free phage population.

Indeed, adsorption rate and lysis time impact both the effective and actual population sizes in the same way (i.e. $N_e/N_T \approx$ const.). By contrast, larger burst sizes increase the effective population size less than the actual population size (Fig 2), resulting in a decrease of $N_e/N_T$. This can be interpreted by noticing that while increasing burst size results in more phage, the number of phage that can actually contribute to the next generation (i.e. the effective population size) is limited by the number of bacteria that are available. Therefore, as burst size is increased, a larger fraction of phage become wasted.

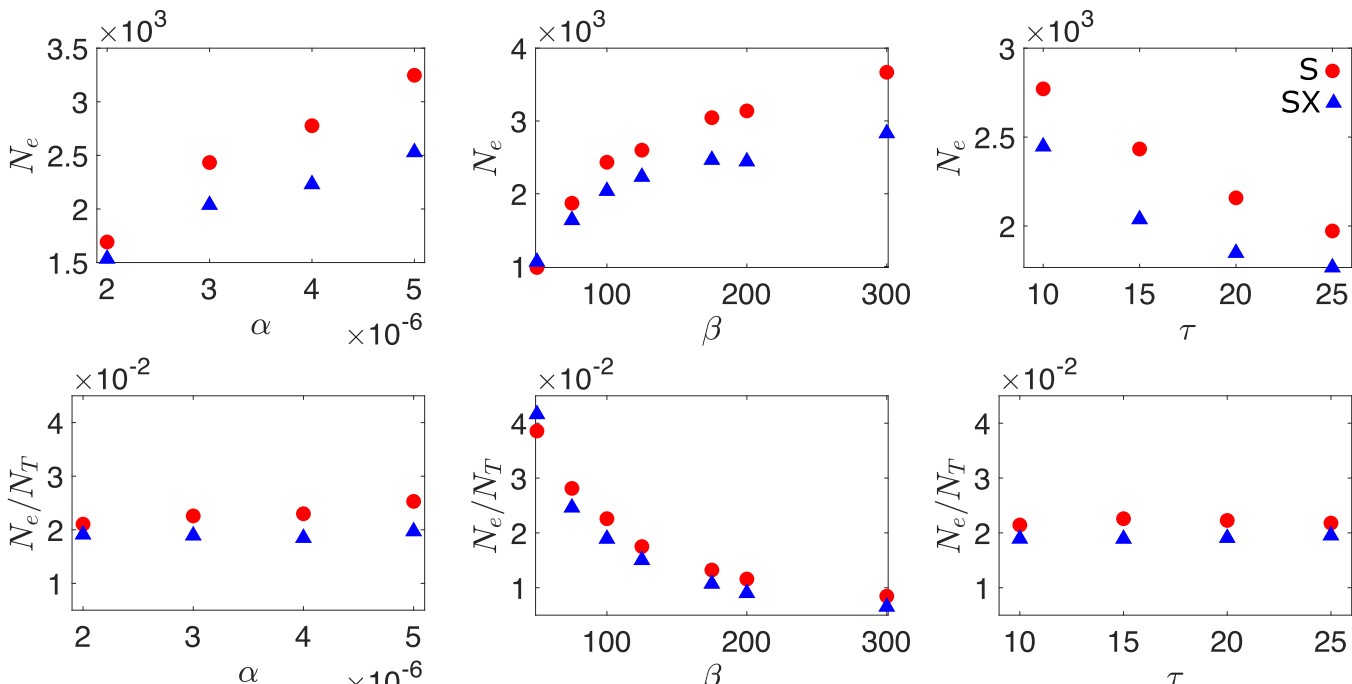

**Fig 2. Effective population size.** The effective population size in both superinfecting (S) and superinfection-excluding (SX) populations as a function of adsorption rate $\alpha$, burst size $\beta$ and lysis time $\tau$. Effective population size are also shown scaled by the size of the total phage population $N_T = (V_{ss} + \beta I_{ss})$. Parameters used were $\alpha = 3 \times 10^{-6}$, $\beta = 100$ and $\tau = 15$ unless otherwise stated. Throughout, $\delta = 0.1$ and $B_0 = 1000$. Error bars are plotted but are too small to see. The data is obtained from an average of at least 1000 independent simulations.

## Neutral mutants are consistently more likely to fix in superinfecting populations

To continue our characterisation of the neutral dynamics in both superinfecting and superinfection-excluding populations, we turn to the fixation probabilities of neutral mutants, and determine how they depend on the phage infection parameters.

Because the total phage population size depends on the life history parameters, the initial mutant frequency corresponding to one mutant phage inoculated in the population also varies with life history parameters. To account for this effect, we re-scale the fixation probability by the initial frequency of the mutant $f_0^* = 1/(V_{ss} + \beta I_{ss})$, which is the same in superinfecting and superinfection-excluding populations. Fig 3 shows that $P_{fix}/f_0^* \approx 1$ as the parameters are varied, indicating that the total number of phages for a given set of parameters is the main controller of neutral dynamics. Indeed, we find that the impact of the life history parameters on the probability of fixation is what one would intuitively expect (S2 Appendix): larger adsorption rate and burst size, and shorter lysis time, increase the steady-state size of the phage population, and reduce $P_{fix}$. By describing the average behaviour of our simulations with a system of ordinary differential equations (ODEs), we confirm that the ODE solution for the total phage population at steady-state $N_T$ is the same as in the stochastic model (S2 Appendix).

Fig 3 also shows that, on average, neutral mutants in the superinfecting scenario are more likely to fix than mutants in an equivalent superinfection-excluding population (blue and red dashed lines in Fig 3 respectively). This result agrees with that found by Wodarz *et al.* [48], who showed that in a superinfecting viral population, higher multiplicities of infection slightly favoured rare neutral and disadvantageous mutants in the short term. The intuition behind

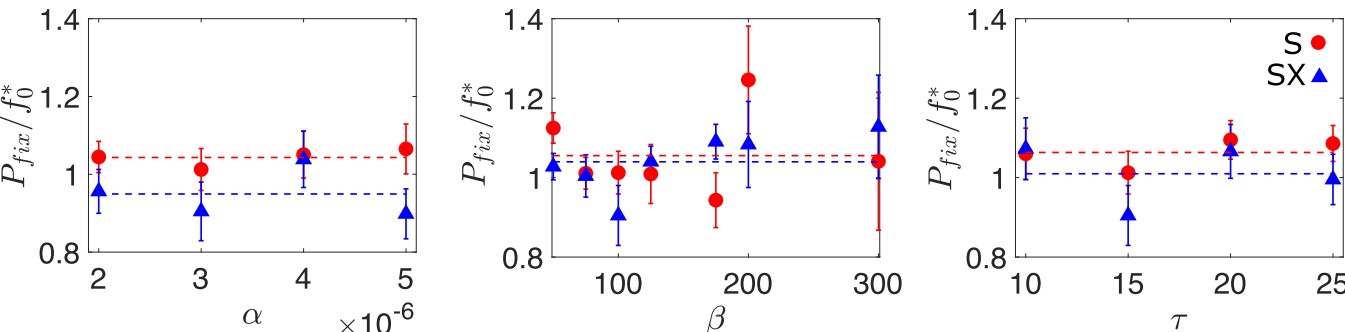

**Fig 3. Fixation of neutral mutants.** Probability of mutant fixation $P_{fix}$ in the superinfecting (S) and non superinfection excluding (SX) scenarios, scaled by the initial frequency of the mutant $f_0^* = 1/(V_{ss} + \beta I_{ss})$, as a function of adsorption rate $\alpha$, burst size $\beta$ and lysis time $\tau$. Dashed lines indicate the average of the data for both the superinfecting (blue) and superinfection-excluding (red) scenarios. These lines indicate that neutral mutants in superinfecting populations experience a small advantage over mutants in an equivalent superinfection-excluding population. Unscaled $P_{fix}$ data can be seen in S2 Appendix. Unless otherwise stated, the parameters used were $\alpha = 3 \times 10^{-6}$, $\beta = 100$, $\tau = 15$, $\delta = 0.1$ and $B_0 = 1000$. The error in our estimate of the fixation probability $\Delta P_{fix}$ is given by $\Delta P_{fix} = \sqrt{n_{fix}}/n$, where $n$ and $n_{fix}$ represent the total number of simulations and the number of simulations where the mutant fixes respectively. The data is obtained from a minimum of 14 million independent simulations.

this observation can be explained in the following way: at the moment that the mutant is introduced, all infected cells are infected by the resident phage. In the superinfecting scenario, the mutant population can therefore grow by infecting an uninfected cell, or by infecting an already infected cell, as this secondary infection will lead to some fraction of the burst size being allocated to the mutant type. While resident phage can replicate by infecting either types of host, the resident population cannot further grow by infecting previously infected cells. This is because all infected cells are already exclusively infected with resident phage, and superinfection of resident infected cells by more resident phage does not result in any more resident phage being produced. As a result, superinfection increases the mutant's chance of survival in the early stages in comparison to the superinfection-excluding counterpart, similarly to conditions of high vs. low MOI [48].

## Higher growth rate does not translate into competitive advantage

To investigate the evolutionary fate of non-neutral mutations, we first characterise how phage growth rate and competitive fitness is affected by changes to the phage life history parameters, i.e., adsorption rate $\alpha$, burst size $\beta$ and the lysis time $\tau$, relative to the values used in our neutral simulations (Fig 3).

S2 Fig shows that increasing burst size or adsorption rate results in a larger selective advantage both in isolation and in direct competition (see Methods). However, while variations in burst size affect similarly the phage growth rate in isolation and its (dis)advantage in a competitive setting ($s_{growth} \approx s_{comp}$, Fig 4), variations in adsorption rate lead to a stronger competitive (dis)advantage than what would be predicted by the growth rate ($|s_{growth}| < |s_{comp}|$). The intuition behind this result is that increasing adsorption rate becomes particularly advantageous in a competitive environment, as being the *first* virus to infect a host allows the virus to have largely (superinfection scenario) or completely (superinfection exclusion scenario) exclusive access to the host resources.

The impact of altering lysis time $\tau$ is surprising. S2 Fig shows that increasing $\tau$ results in a reduced growth rate, as intuition suggests. Yet, in the superinfection-excluding scenario no discernible impact on $s_{comp}$ is observed (Fig 4). This result is supported by our ODE model (S2 Appendix), which shows that once the system is at steady-state, alterations to lysis time offer no advantage to one phage over the other (S3 Fig). We believe that this is a special feature of a

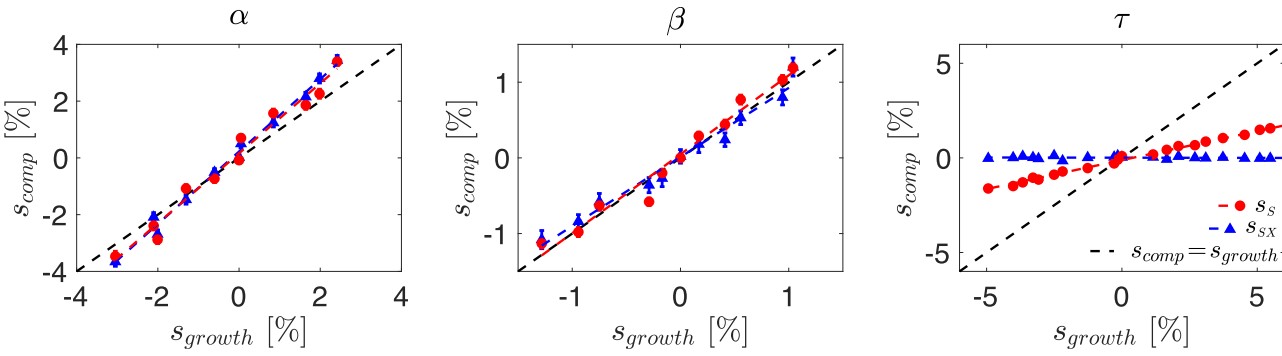

**Fig 4. Competitive vs isolated selective advantage.** The selective advantage in a competitive setting $s_{comp}$ as a function of the change in growth rate $s_{growth}$, when changing adsorption rate $\alpha$, burst size $\beta$ and lysis time $\tau$. Straight line fits are shown as dashed lines, with gradient $\sigma$ such that $s_{comp} = \sigma s_{growth}$. From the above data we find $\sigma_{S\alpha} = 1.2324$, $\sigma_{SX\alpha} = 1.2764$, $\sigma_{S\beta} = 1.0432$, $\sigma_{SX\beta} = 0.9134$, $\sigma_{S\tau} = 0.3057$ and $\sigma_{SX\tau} \approx 0$. Resident parameters used were $\alpha = 3 \times 10^{-6}$, $\beta = 100$ and $\tau = 15$. As before $\delta = 0.1$ and $B_0 = 1000$. $s_{growth}$ determined from 500 simulations, and $s_{comp}$ determined from 200 simulations. Error bars are given by the standard error on the mean of the simulations. Error bars on x axis have been omitted for clarity, but are shown in S2 Fig.

turbidostat setting, as lysed hosts are immediately replaced by uninfected cells, providing the same number of viable hosts independently of the time needed by the phage to lyse them. By contrast, in the superinfecting case, we are able to observe a selective (dis)advantage in direct competition, although at a significantly reduced level compared to the change in growth rate. We believe that this effect arises because, while the extracellular competition is limited by the turbidostat setup, in the superinfecting scenario there is the opportunity for some intracellular competition to occur, as mutants will grow at different rates inside the host, resulting in different numbers of phage (both in total and proportionally) being released upon lysis. We leave a full characterisation of the relationship between growth rate in isolation and competitive fitness to future works.

## Superinfection results in more efficient selection

Having characterised how changes to the phage infection parameters alter first genetic drift and second fitness, we now put both ingredients together and investigate the dynamics of non-neutral mutants. To this end, we simulate a resident phage population to steady state, introduce a single non-neutral mutant and then run the simulation until extinction or fixation occurs.

In agreement with our observations regarding the difference between growth rate and competitive fitness, we find that the value of $s_{growth}$ is not sufficient to determine the fixation probability of the corresponding mutant (Fig 5): a mutation associated with a higher adsorption rate $\alpha$ increases the mutant's chance to fix more than a mutation which alters the burst size $\beta$ and leads to the same growth rate. We also find that beneficial mutations are consistently more likely to fix (and deleterious mutations more likely to go extinct) in superinfecting populations (red) than superinfection-excluding populations (blue). This suggests that superinfection improves selection efficiency, by more readily fixing beneficial mutations and purging deleterious ones.

To provide a theoretical framework to our findings, we compare the simulation data to the fixation probabilities one would expect in a corresponding Moran model. For small selective advantage $s_{comp}$, the probability of fixation is given by

$$P_{fix} = \frac{1 - e^{-N_e s_{comp} f_0}}{1 - e^{-N_e s_{comp}}}, \tag{1}$$

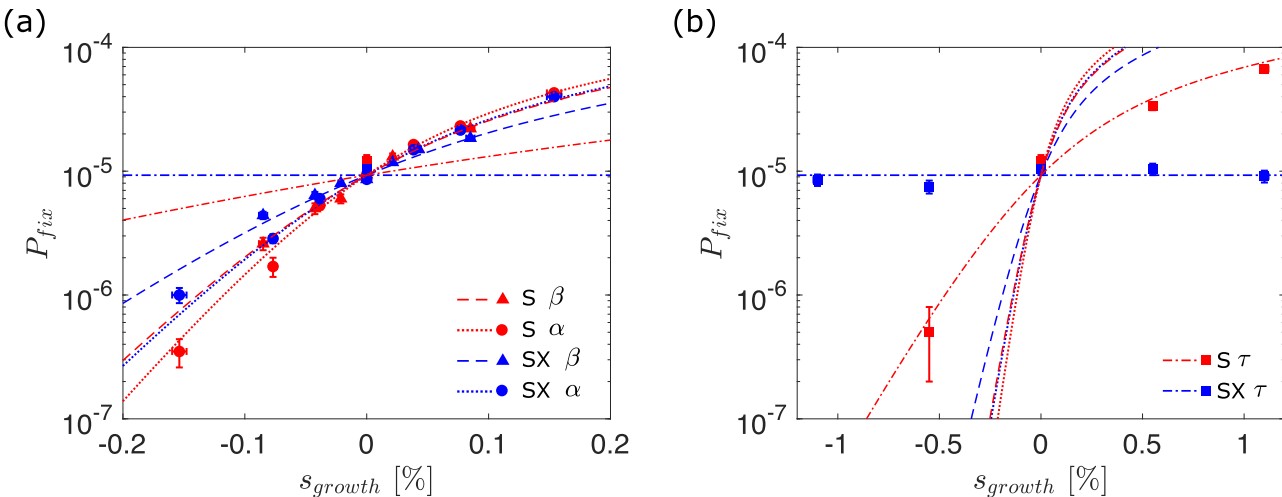

**Fig 5. Fixation of non-neutral mutants.** Probability of mutant fixation $P_{fix}$ as a function of selective growth advantage $s_{growth}$. Points indicate simulation results, while lines indicate theoretically predicted values in a Moran model with equivalent parameters (Eq 1). Data points for the $\alpha$ and $\beta$ mutants have been omitted from the right hand panel for clarity. The error in our estimate of the fixation probability $\Delta P_{fix}$ is given by $\Delta P_{fix} = \sqrt{n_{fix}}/n$, where $n$ and $n_{fix}$ represent the total number of simulations and the number of simulations where the mutant fixes respectively. Error bars in the $x$-axis represent the errors on the growth rate fitness $s_{growth}$ that each burst size corresponds to. These are calculated by fitting a linear relation to growth rate measurements such that $s_{growth} = m(\beta_{mut} - \beta_{res})$. The fractional error on the $s_{growth}$ is then equal to the fractional error on the fitted gradient $m$. The data is obtained from a minimum of 5 million independent simulations.

where $f_0$ is the initial frequency of the mutant in the population with effective population size $N_e$ [52, 53]. Our earlier results on neutral dynamics and fitness provide independent measurements of the parameters in Eq 1 for different values of $\alpha$, $\beta$ and $\tau$: $f_0 = f_0^*$ from our initial condition (i.e., $1/N_T$, where $N_T$ is the steady-state phage population size when the mutant is introduced); $N_e$ is measured from the decay of heterozygosity (Fig 2); and $s_{comp} = \sigma s_{growth}$ is derived from our measurements of the relationship between competitive and growth rate advantage (Fig 4). These theoretical predictions are plotted without additional fitting parameters as lines in Fig 5.

Fig 5 shows that the theoretical prediction from the appropriately parameterised Moran model matches the simulation data remarkably well, despite the complex internal infection dynamic (see S3 Appendix for quantitative comparison). We note, however, that the simulation data consistently fails to intersect at the same point when $s_{growth} = 0$ in the superinfecting scenario. This is because of the effect outlined in Fig 3, where rare mutants initially experience a slight advantage in the superinfecting scenario because they are able to increase in number by infecting both uninfected and infected cells. To test the validity of our findings across parameter space, we also perform all of the above analysis with different resident parameters, obtaining similar results (S4 Appendix).

## Superinfection exclusion slows down adaptability in the long run, but is a winning strategy in the short term

Our findings imply that, even in the absence of intra-cellular processes such as recombination, superinfection results in more efficient selection, so that beneficial mutations are relatively more likely to fix, and deleterious ones are more likely to be purged, leading to a fitter overall population in the long run. From the point of view of viral adaptation, allowing superinfection ultimately seems like the better long-term strategy. It is therefore puzzling why several natural phage populations have developed sophisticated mechanisms to prevent superinfection,

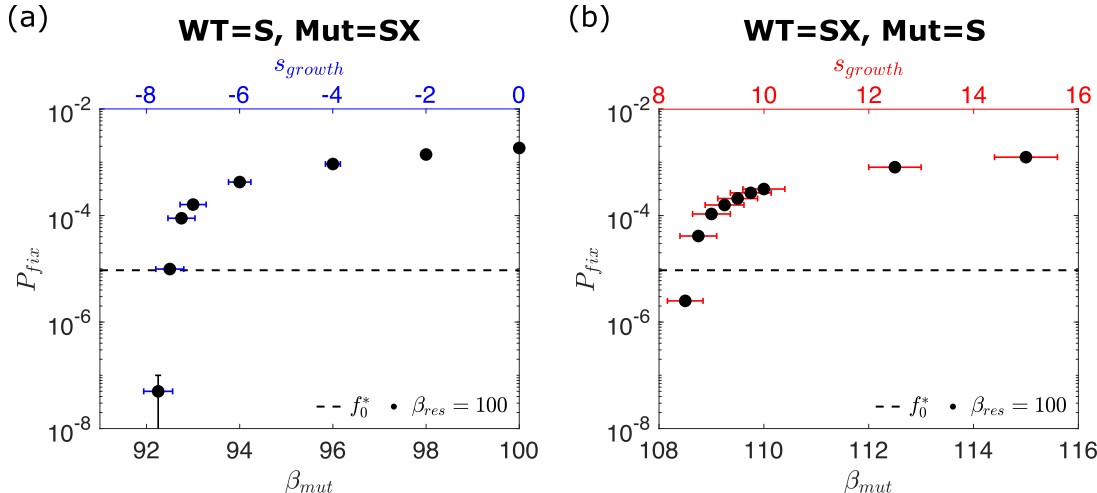

**Fig 6. Mutations which alter the ability to prevent superinfection.** (a) The probability $P_{fix}$ of a mutant which prevents superinfection fixing in a population that allows it, as a function of mutant burst size $\beta_{mut}$. (b) The probability $P_{fix}$ of a mutant which allows superinfection fixing in a population that prevents it, as a function of mutant burst size $\beta_{mut}$. It can be seen that the superinfecting mutant requires a significantly increased burst size to fix, and conversely the superinfection-excluding mutant can fix, even if its burst size is greatly reduced. The error in our estimate of the fixation probability $\Delta P_{fix}$ is given by $\Delta P_{fix} = \sqrt{n_{fix}}/n$, where $n$ and $n_{fix}$ represent the total number of simulations and the number of simulations where the mutant fixes respectively. Error bars in the $x$-axis represent the errors on the growth rate fitness $s_{growth}$ that each burst size corresponds to. These are calculated by fitting a linear relation to growth rate measurements such that $s_{growth} = m(\beta_{mut} - \beta_{res})$. The fractional error on the $s_{growth}$ is then equal to the fractional error on the fitted gradient $m$. The fixation data is obtained from a minimum of 20 million independent simulations.

particularly given that employing these mechanisms is expected to come with a biological cost, such as reduced burst size [54, 55] or increased lysis time [56].

To address this question, we consider the fate of mutations that either (i) remove the mutant's ability to prevent superinfection in a resident superinfection-excluding population or (ii) provide the mutant the ability to prevent superinfection in a resident superinfecting population. Fig 6 shows that if the mutant is neutral ($\beta_{mut} = \beta_{res} = 100$), then the superinfection-excluding mutant is two orders of magnitude more likely to fix than the expectation based on its initial frequency $f_0^*$, and that, by contrast, the superinfecting mutant is at least two orders of magnitude more likely to go extinct. It should be noted that we actually find no instances of mutant fixation in this case, but our detection power is limited by the number of simulation runs. Here, we run at least 20 million simulations, and we can thus infer that $P_{fix} \ll 10^{-7}$. This indicates that mutants which are able to prevent superinfection experience a very strong selective advantage over their superinfecting counterparts, and vice-versa.

To account for the possibility that superinfection exclusion comes at a cost in phage growth, as preventing superinfection likely requires the production of extra proteins, the resources for which could otherwise have gone to the production of more phage, we consider the case where superinfection exclusion is associated with a reduction in burst size [54]. Remarkably, we find that even when preventing superinfection carries a burden of 7% reduction in burst size ($s_{growth} < -7\%$), the superinfection-excluding mutant still fixes more often than a neutral superinfecting mutant (Fig 6). Conversely, a minimum of 8% increase in burst size ($s_{growth} > 8\%$) is necessary to give a superinfecting mutant any chance of fixing in a superinfection-excluding population. This indicates that while allowing superinfection increases selection efficiency at the population level, preventing it is ultimately a winning strategy in the short term, partially explaining why superinfection exclusion is so common in nature [5, 6].

## Discussion

In this work, we have considered the impact of either allowing or preventing superinfection on the evolution of viral populations. Using a stochastic agent-based model of viral infection, we have shown that allowing superinfection reduces the strength of genetic drift, leading to an increase in effective population size. Weaker fluctuations result in a higher efficiency of selection in viral populations, with beneficial mutations fixing more frequently, and deleterious ones more readily being purged from the population. Despite the long term, population-wide benefit of allowing superinfection, we find that if a mutant arises which is capable of preventing superinfection, it will fix remarkably easily, even if its growth rate is heavily compromised. Conversely, if the whole population is capable of preventing superinfection, mutants which allow it will have almost no chance of ever succeeding.

The evolutionary impact of superinfection (and more generally multiple infections) has most often focused on the role of intracellular interactions and competition [11–14, 16–19], such as genetic recombination and reassortment [20–23], and viral complementation [23–27]. A prevalent finding (amongst others) is that recombination and reassortment can improve the efficiency of selection in viral populations which do not exclude superinfection. Remarkably, our work demonstrates that the basic occurrence of superinfection alone, absent of any recombination or reassortment, is capable of increasing the selection efficiency. In this context, our results provide a useful baseline for comparison when trying to assess the significance of each of these more complex effects, which may or may not be present in different situations.

An unexpected finding of this work is that in the turbidostat system we consider, while increased adsorption rate and burst size both increase the fitness of the phage population in all respects, in the superinfecting scenario lysis time plays a significantly reduced role in the competitive (dis)advantage experienced once the system has reached a steady-state, and in the superinfection-excluding scenario it plays no role whatsoever. While it has been demonstrated previously that changes to fecundity and generation time can have different impacts on mutation fixation probability, even when they have the same impact on long-term growth rate [57], our result is somewhat in contrast with previous studies showing that well-mixed liquid cultures with an abundance of hosts generally select for higher adsorption rates and lower lysis times [51, 58–60]. The key difference between such liquid cultures and the turbidostat system we model here is that in the former host cells are not maintained at a constant density, but the phage population continues to grow until no bacteria remain. This finding illustrates how the presence or absence of a co-existing steady-state between phage and bacteria completely alters the selective pressure on the phage with important implications for studies into the co-evolution of phage and bacteria populations using continuous culturing set-ups [61–63]. In particular, our results suggest that in an evolutionary experiment in a turbidostat, the virus should evolve towards very large burst size even if this feature comes at the cost of longer lysis times, especially if superinfection exclusion occurs [59]. Reciprocally, detecting a selective pressure on lysis time could be used to identify potential phages that allow superinfection, as, in this case, a shorter lysis time is slightly advantageous all else being equal.

Following this, it is natural to wonder how the (dis)advantages and impact of either strategy depends on the selective pressure experienced in different environments. The relationship between viral fitness and the phage life-history parameters (adsorption rate, lysis time and burst size) has been shown to be very context-dependent in both well-mixed and spatially structured settings. For instance, as noted previously, well-mixed settings generally favour higher adsorption rates [64], but in spatially structured settings phage with lower adsorption rates are more successful [65, 66]. Additionally, it has been shown previously that eco-evolutionary feedbacks at the edge of expanding viral populations can result in travelling waves with

vastly different effective population sizes [67]. Given that competition for resources (i.e. viable hosts) in spatially structured environments is local rather than global, phage are more likely to be in competition with other genetically identical phage released by nearby cells. It is therefore possible that superinfection exclusion proves less useful in this context than in well-mixed environments where competition is global and phage are more likely to encounter other genetically different viruses. All of this points at the role of superinfection strategies and other social viral behaviour on the eco-evolutionary dynamics of spatially expanding viral populations as an exciting avenue for future research.

## Methods

### Measuring effective population size of the phage population

Consistently with previous work [52], we expect that the neutral standing diversity of the phage population, quantified by the heterozygosity $H$, will decay exponentially at long times due to genetic drift, so that $H(t) \propto e^{-2t/N_e}$ (S4 Fig), with the decay rate in units of generations being expressed in terms of an effective population size $2/N_e$ (Moran model [52]).

We track the viral heterozygosity $H$ as a function of time, which in a biallelic viral population is given by

$$H = 2\langle f(1-f) \rangle, \tag{2}$$

where $f$ and $(1 - f)$ represent the frequencies of two neutral viral alleles in the population, and $\langle \ldots \rangle$ indicates the average over independent simulations. $H(t)$ can be understood to be the time-dependent probability that two individuals chosen from the population are genetically distinct.

To determine the generation time $T$, we first calculate the *net reproduction rate $R_0$*, which represents the number of offspring an individual would be expected to produce if it passed through its lifetime conforming to the age-specific fertility and mortality rates of the population at a given time (i.e. taking into account the fact that some individuals die before reproducing) [68]. $R_0$ can be calculated as

$$R_0 = \sum l_t m_t, \tag{3}$$

where $l_t$ represents the proportion of individuals (in our case, phage) surviving to age $t$, and $m_t$ represents the average number of offspring produced at age $t$.

There are two mechanisms in our simulations by which phages can 'die' when superinfection exclusion applies: either by decaying with rate $\delta$, or by adsorbing to an infected host with rate $\alpha I_{ss}$. In a sufficiently small timestep $\Delta t$, these rates correspond to a proportion $\delta \Delta t$ and $\alpha I_{ss} \Delta t$ of the total phage, respectively. Equivalently, these can be considered to be the probability that any single phage will die in the same period. As a result, the probability of a phage surviving to age $t$ is $l_t = (1 - \delta \Delta t - \alpha I_{ss} \Delta t)^{t/\Delta t}$.

The average number of offspring $m_t$ produced at age $t$ is 0 if $t < \tau$, because we assume that no phage is released before the lysis time. For $t > \tau$, $m_t$ is given by the probability of successfully infecting a viable host in a timestep $\Delta t$, $\tau$ time earlier ($\alpha B_{ss} \Delta t$), multiplied by the yield of new phage ($\beta - 1$).

In the limit where $\Delta t \to 0$, this will result in a net reproductive rate of the form

$$R_0 = \lim_{\Delta t \to 0} \sum_{t=0}^{\infty} m_t l_t = \lim_{\Delta t \to 0} \sum_{t=\tau}^{\infty} \Delta t \alpha B_{ss} (\beta - 1)(1 - \Delta t(\delta + \alpha I_{ss}))^{t/\Delta t}, \tag{4}$$

$$= \int_{t=\tau}^{\infty} \alpha B_{ss}(\beta - 1)e^{-(\delta + \alpha I_{ss})t}\mathrm{d}t, \tag{5}$$

$$= \frac{\alpha B_{ss}(\beta - 1)}{\delta + \alpha I_{ss}}e^{-(\delta + \alpha I_{ss})\tau}, \tag{6}$$

where the integral starts at $\tau$ because no offspring are produced prior to that point.

Then the generation time $T$, defined as the average interval between the birth of an individual and the birth of its offspring, is

$$T = \lim_{\Delta t \to 0} \frac{\sum tl_t m_t}{R_0} = \frac{\int_{t=\tau}^{\infty} t\alpha B_{ss}(\beta - 1)e^{-(\delta + \alpha I_{ss})t}\mathrm{d}t}{R_0} = \tau + \frac{1}{\delta + \alpha I_{ss}}. \tag{7}$$

Here, we will use resident phage parameters $\alpha = 3 \times 10^{-6}$, $\tau = 15$, $\delta = 0.1$ and a total bacterial population of $B_0 = 1000$, which leads to $I_{ss} = 681$ and a generation time of $T = 24.8$. This generation time is also supported by stochastic simulations of the phage adsorption and death processes (S5 Appendix). Throughout this work, we use the same generation time for both superinfecting and superinfection-excluding populations (more details in S5 Appendix).

For comparison, coliphage T7 in liquid culture typically has parameters of $\tau \approx 10 - 20$ min, $\alpha \approx 10^{-9}$ ml/min and $B_0 \approx 10^6 - 10^8$ ml$^{-1}$, thereby yielding an $\alpha B_0 \approx 10^{-3} - 10^{-1}$ min$^{-1}$ [59, 69]. These values are comparable to our own if we equate 1 timestep = 1 min, and so $\tau = 15$ min and $\alpha B_0 = 3 \times 10^{-3}$ min$^{-1}$, such that the relative timescales in our simulation remain consistent. The reason behind choosing a larger adsorption rate and smaller bacteria population is purely practical, as the alternative would lead to unreasonably long computational times. Given these values, our choice of decay rate $\delta$ is made such that steady-state population sizes are reached.

## Measuring mutant fitness and growth rate

We start by defining a selective advantage $s_{growth}$ in terms of the exponential growth rate $r_{mut}$ of the mutant phage population relative to that of the resident phage $r_{res}$ [70]:

$$s_{growth} = \frac{r_{mut}}{r_{res}} - 1. \tag{8}$$

The exponential growth rate is determined by simulating the growth of the corresponding phage population in isolation, and performing a linear fit to the log-transformed phage number as a function of time, which is then averaged over 500 independent simulations. It should be noted that as there is only one type of phage in these simulations, the growth rate of both superinfecting and superinfection-excluding populations is the same.

We also characterised the fitness of mutants in a competitive setting, by simulating a resident population until steady state, and then replacing 50% of the population with the mutant. In this direct competition scenario, we determine the selective (dis)advantage $s_{comp}$ of the mutant phage by tracking the relative growth of mutant and resident populations, so that

$$\frac{V_{mut}}{V_{res}} = \frac{V_{mut}(t=0)e^{r_{res}(1+s_{comp})t}}{V_{res}(t=0)e^{r_{res}t}} = e^{r_{res}s_{comp}t}, \tag{9}$$

as $V_{mut}(t=0) = V_{res}(t=0)$. $s_{comp}$ is determined from the average of 200 simulations. Importantly, in contrast to $s_{growth}$, this competitive selective advantage ($s_{comp}$) can in principle differ between superinfecting ($s_S$) and superinfection-excluding ($s_{SX}$) phage populations. In the

absence of any interactions between the two competing phage populations, $s_{growth}$ and $s_{comp}$ are typically expected to be the same.

## Measuring mutant probability of fixation

To measure fixation probabilities of individual mutations, we allow our simulations to reach steady state, we then introduce a single mutant phage into the free phage population, and run the simulation until either fixation or extinction occurs. This process is repeated at least 5 million times for each set of parameters. The probability of mutant fixation $P_{fix}$ is determined from the fraction of simulations where the mutant fixed, $n_{fix}$, over the total number of simulations run, $n$ (i.e. $P_{fix} = n_{fix}/n$). Assuming a binomial distribution, the error in our estimate of the number of fixation events $\Delta n_{fix}$ is given by $\Delta n_{fix} = \sqrt{nP_{fix}(1 - P_{fix})}$. Consequently, our error in the estimate of fixation probability $\Delta P_{fix}$ is given by $\Delta P_{fix} = \sqrt{P_{fix}(1 - P_{fix})/n}$. It can be easily verified that in the case where $n_{fix} \ll n$, as we have here, the error approaches $\Delta P_{fix} = \sqrt{n_{fix}}/n$ as would be found in a Poisson distribution.

## Supporting information

**S1 Fig. Steady-states are independent of intial conditions.** The steady-state phage population $V_{ss}$ reached does not depend on the initial number of phage $V_0$ in the simulations. In all, $\alpha = 3 \times 10^{-6}$, $\beta = 100$, $\tau = 15$, $\delta = 0.1$ and $B_0 = 1000$.
(EPS)

**S2 Fig. $s$ as a function of phage life-history parameters.** The selective advantage $s$ relative to a resident phage that results from a change to adsorption rate $\alpha$, burst size $\beta$ and lysis time $\tau$. This is measured both in terms of the effect on the isolated growth rate of the mutant ($s_{growth}$, Eq 8), and in terms of the change in frequency in a population initiated with 50% mutant and 50% resident ($s_{SX}$ and $s_S$, Eq 9). Resident parameters used were $\alpha = 3 \times 10^{-6}$, $\beta = 100$ and $\tau = 15$. As before $\delta = 0.1$ and $B_0 = 1000$. $s_{growth}$ determined from 500 simulations, and $s_{comp}$ determined from 200 simulations. Error bars are given by the standard error on the mean of the simulations.
(EPS)

**S3 Fig. $s_{comp}$ in the ODE model.** The relative change in frequency of two populations in the ODE model (indicating the average behaviour in the stochastic model). It can be seen that once at steady-state, changing lysis time $\tau$ has no effect. Parameters used were $\alpha = 3 \times 10^{-6}$, $\beta = 100$ and $\tau = 15$ unless otherwise stated. As throughout, $\delta = 0.1$ and $B_0 = 1000$.
(EPS)

**S4 Fig. Example decay in heterozygosity.** Linear fit to log transformed heterozygosity data, with slope $\Lambda \equiv 2/N_e$ revealing that allowing superinfection (red) results in a larger effective population size compared to the case where superinfection is prevented (blue). Parameters used were $\alpha = 3 \times 10^{-6}$, $\beta = 100$, $\tau = 15$, $\delta = 0.1$ and $B_0 = 1000$. Data obtained is the average of 1000 independent simulations.
(EPS)

**S1 Appendix. Stochasticity in lysis time.** Here we discuss the decision to not incorporate stochasticity in lysis time in the model presented in the main text.
(PDF)

**S2 Appendix. ODE description of model.** The average behaviour of the model used in the main text is described by a set of ordinary differential equations (ODEs), showing good agreement with our stochastic simulations.
(PDF)

**S3 Appendix. Comparison with expectation from Moran model.** A quantitative comparison between the fixation probabilities obtained in our stochastic simulations with those that would be predicted in a similarly parameterised Moran model.
(PDF)

**S4 Appendix. Repeat measurements with $\beta_{res}$ = 70.** Here we repeat a subset of the measurements carried out in the main text with different resident phage parameters, in this instance $\beta_{res}$ = 70.
(PDF)

**S5 Appendix. Calculation of generation time.** Here we support the generation time calculated in the main text with results of stochastic simulations. We also include a more detailed discussion about the differences in generation time between superinfecting and superinfection-excluding populations.
(PDF)

## Author Contributions

**Conceptualization:** Michael Hunter, Diana Fusco.

**Formal analysis:** Michael Hunter.

**Investigation:** Michael Hunter.

**Methodology:** Michael Hunter, Diana Fusco.

**Project administration:** Diana Fusco.

**Resources:** Diana Fusco.

**Software:** Michael Hunter.

**Supervision:** Diana Fusco.

**Validation:** Michael Hunter.

**Visualization:** Michael Hunter.

**Writing – original draft:** Michael Hunter.

**Writing – review & editing:** Michael Hunter, Diana Fusco.

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
