## [Decision Letter · Decision Letter 0]

21 Jan 2022

Dear Mr. Hunter,

Thank you very much for submitting your manuscript "Superinfection exclusion: a viral strategy with short-term benefits and long-term drawbacks" for consideration at PLOS Computational Biology.

As with all papers reviewed by the journal, your manuscript was reviewed by members of the editorial board and by several independent reviewers. In light of the reviews (below this email), we would like to invite the resubmission of a significantly-revised version that takes into account the reviewers' comments.

The reviewers are broadly enthusiastic about this work, though Reviewer 2 raises an important issue about the effect of certain assumptions upon the validity of the results. This issue, alongside other issues raised by both reviewers, should be dealt with in a revised version of the manuscript.

We cannot make any decision about publication until we have seen the revised manuscript and your response to the reviewers' comments. Your revised manuscript is also likely to be sent to reviewers for further evaluation.

Sincerely,

Christopher Illingworth

Guest Editor

PLOS Computational Biology

Ville Mustonen

Deputy Editor

PLOS Computational Biology

The reviewers are broadly enthusiastic about this work, though Reviewer 2 raises an important issue about the effect of certain assumptions upon the validity of the results. This issue, alongside other issues raised by both reviewers, should be dealt with in a revised version of the manuscript.

Reviewer's Responses to Questions

**Comments to the Authors:**

Reviewer #1: Overall, I really enjoyed this paper. The question of whether superinfection exclusion poses advantages or disadvantages to the evolution of phage populations is a longstanding one, but the authors manage to present their approach in a way that appears straightforward while also making a nice contribution to the field. I find that their results justify their conclusions, though I have several minor points that I think could be addressed to make the work even more clear. There were a few areas where I think additional exposition would help the reader understand the authors’ points.

I found the introduction well-written and it poses the questions the authors aim to address very clearly. I think that the breadth of literature covered in the introduction does a good job of summarizing the gaps of knowledge in this sub-field and where progress has been made so far. I would also recommend citing the sociovirology review written by Sam Diaz-Munoz, Rafael Sanjuan, and Stuart West (10.1016/j.chom.2017.09.012), which is in my view one of the more important pieces written on the topic.

Lines 85-86: The authors state that adsorption that happens at times later than the first adsorption event do not contribute DNA to the host cell in the superinfection excluding scenarios. Does this mean that the phages that enter later are removed from the free phage population, but do not get replicated? This would be in contrast, I think, to what is described in the Superinfecting section that begins on line 126 where subsequent adsorption events increase the within-cell population until lysis. Lines 123 to 125 may be a good place to clarify.

Lines 121-123: The lysis time is not drawn from a distribution with parameter tau, unlike burst size which has mean beta, or the number of phage that adsorb in a timestep which depends on alpha. Why lysis time would not be stochastic, but burst size and adsorption are stochastic, is not clear to me so I would appreciate it if the authors motivated this choice.

Lines 194-197: The juxtaposition here is throwing me. Why would a neutral mutant have two cell types available to it while the wild-type can only infect uninfected cells? Is it because we are at steady state when the mutant infects, so it has multiple cell-states available, whereas the wild-type before steady state only expanded through uninfected cells? Please elaborate.

Line 206: You introduce parameter s here, but I don’t think it was defined in the methods. Is it just the general term for s_growth and s_comp? If so, please state.

Line 252: How do sigma and phi relate to each other? You describe them as fitting and scaling parameters for the various s variables, but their roles are unclear.

Figure 6: It would be nice if you listed the number of simulations ran for each of your figures and their subplots. It is in the methods for some of the work but I don’t think it encompasses all of the figures. looks like 1/10^6.5 based on 6b if the leftmost point has been adjusted given your absence of observed fixation. I do appreciate that they state the limitation that they did not observe fixation here, though would the value plotted be an upper bound instead of a lower bound like the authors write? More simulations to observe a rare fixation event would decrease the estimated probability, no?

Figure 6: The error bars need to be clarified. I assume it has something to do with s_growth reduction but it really isn’t clear to me how the bars relate to the points as they seem to be varying different parameters, but wouldn’t an additional dimension need to be displayed to adequately show the exploration of this parameter space?

Lines 328-330: I think this is an excellent point.

Line 404: Is this alpha value consistent with what is explored in the figures? Here you state that the empirical value is on the order of 10^-9, but the parameters explored in the figures are on the order of 10^-6.

I thought that the Methods section was very well written – nice work. Particularly the section on calculating the fixation probability.

I also found several spelling errors, so I think the authors should give a careful read if they resubmit. Examples:

Figure 1 caption: “superinfecting in permitted” should instead be “is permitted”

Line 332: “our results suggests” should be “suggest”

Reviewer #2: The review is uploaded as an attachment.

**Have the authors made all data and (if applicable) computational code underlying the findings in their manuscript fully available?**

Reviewer #1: Yes

Reviewer #2: **No: **The authors provide a link to their code in the manuscript, but for some reason it cannot be accessed.

PLOS authors have the option to publish the peer review history of their article (what does this mean?). If published, this will include your full peer review and any attached files.

Reviewer #1: No

Reviewer #2: No
---

## [Decision Letter · Decision Letter 1]

20 Apr 2022

Dear Mr. Hunter,

We are pleased to inform you that your manuscript 'Superinfection exclusion: a viral strategy with short-term benefits and long-term drawbacks' has been provisionally accepted for publication in PLOS Computational Biology.

Please note that your manuscript will not be scheduled for publication until you have made the required changes, so a swift response is appreciated.  We recommend that further changes be considered in line with the suggestions of reviewer 2.

Best regards,

Christopher Illingworth

Guest Editor

PLOS Computational Biology

Ville Mustonen

Deputy Editor

PLOS Computational Biology

Your manuscript has been considered again by the reviewers, who are happy that your work merits publication in PLoS Computational Biology. In the preparation of a final manuscript we recommend consideration be given to the thoughts raised by reviewer 2, but we are happy for changes to be made at the authors' discretion.

Reviewer's Responses to Questions

**Comments to the Authors:**

Reviewer #2: Hunter and Fusco have thoroughly revised their manuscript in accordance with the issues and suggestions raised by reviewer 1 and myself. At least in my case, in this new version of the paper, the authors have resolved most of my concerns. The issues regarding the absence of intracellular competence in the previous computational framework have been addressed as I suggested, except that the time to the cell lysis is still defined by the first infecting phage. However, I understand that this is probably the computationally simplest approach and I doubt very much that a different setting would qualitatively change the conclusions of the paper.

The authors have also introduced changes in relation to my criticisms about non-necessity of high MOI regimens for superinfection to occur and have introduced a terminology for the use of coinfection and superinfection terms in accordance with the definitions proposed previously by Turner and Duffy 2008. This improves the readability of the article, because although, under the definitions used, any superinfection event can be considered a coinfection event, the opposite is not true, and the inconvenient use of superinfection is avoided when coinfection in a stricter sense (i.e. coinfection without superinfection) would also be valid or even more appropriate (e.g. see lines 47, 51, and 55). In these cases, the term multiple infection has been introduced, which encompasses cases of strict coinfection as well as superinfection. However, by defining the term "multiple infection" within the host (i.e. “any case where multiple viruses exist within a single host simultaneously”, lines 32-34) it is not necessarily confined to the cellular level, although it is in the case of phages, which is the focus of the article.

Overall, because of these and all the other changes, corrections and additional information included in the new manuscript, I think the article is now acceptable to publish. However, I still want to mention two things that have raised some doubts.

(I) As the authors mention in line 236, I find the null effect of tau on fitness in competition (s_comp) in the superinfection exclusion scenario surprising. Although I am confident that the result is correct, the explanation about the nature of the turbidostat system (lines 241-243) does not entirely satisfy me. If both resident and mutant viruses share adsorption rate and burst size, but the mutant lyses cells faster, even though the number of lysed cells is immediately available as susceptible cells, shouldn't this mutant phage become more common in the population as it carries out more infection cycles in the same time frame? By doing do it would be expected to produce a greater number of viral progeny and susceptible cells than the phage with longer tau.

(II) For the calculation of the error (I assume SEM) in estimated fixation probabilities, it is considered that the probability of fixation fits a binomial distribution (lines 454-456). Under this assumption I don't see why error is defined as sqrt(nfix)/n. I would understand that this would be the case if the variance equaled the mean (which is nfix/n). Maybe I am wrong, but I understand that in each simulation can be considered a Bernoulli trial, so the variance in the Pfix parameter should be Pfix(1-Pfix) and the SEM sqrt(Pfix(1-Pfix)/n). Therefore, shouldn't you arrive at an expression, in the terms used in the article, such that sqrt((n*nfix-nfix^2)/n^3))?

**Have the authors made all data and (if applicable) computational code underlying the findings in their manuscript fully available?**

Reviewer #2: Yes

PLOS authors have the option to publish the peer review history of their article (what does this mean?). If published, this will include your full peer review and any attached files.

Reviewer #2: No

---

## [Editor Report · Acceptance letter]

5 May 2022

PCOMPBIOL-D-21-02234R1 

Superinfection exclusion: a viral strategy with short-term benefits and long-term drawbacks

Dear Dr Hunter,

I am pleased to inform you that your manuscript has been formally accepted for publication in PLOS Computational Biology. Your manuscript is now with our production department and you will be notified of the publication date in due course.

With kind regards,

Agnes Pap
